# Various Statistical Approaches to Assess and Predict Carcass and Meat Quality Traits

**DOI:** 10.3390/foods9040525

**Published:** 2020-04-22

**Authors:** Marie-Pierre Ellies-Oury, Jean-François Hocquette, Sghaier Chriki, Alexandre Conanec, Linda Farmer, Marie Chavent, Jérôme Saracco

**Affiliations:** 1Bordeaux Science Agro, 1 cours du Général de Gaulle, CS 40201, 33175 Gradignan, France; alexandre.conanec@agro-bordeaux.fr; 2INRAE, UMR1213 Herbivores, 63122 Saint Genès Champanelle, France; jean-francois.hocquette@inrae.fr; 3Clermont Université, VetAgro Sup, UMR1213 Herbivores, BP 10448, 63000 Clermont-Ferrand, France; 4Isara Agro School for Life, 23 rue Jean Baldassini, 69364 Lyon CEDEX 07, France; schriki@isara.fr; 5Université de Bordeaux, UMR5251, INRIA, 33400 Talence, France; marie.chavent@math.u-bordeaux.fr (M.C.); jerome.saracco@math.u-bordeaux.fr (J.S.); 6Agri-Food and Biosciences Institute, 18a Newforge Lane, Belfast BT9 5PX, UK; Linda.Farmer@afbini.gov.uk

**Keywords:** optimization, meat quality, trade-off, meat standards Australia, carcass, bovine

## Abstract

The beef industry is organized around different stakeholders, each with their own expectations, sometimes antagonistic. This article first outlines these differing perspectives. Then, various optimization models that might integrate all these expectations are described. The final goal is to define practices that could increase value for animal production, carcasses and meat whilst simultaneously meeting the main expectations of the beef industry. Different models previously developed worldwide are proposed here. Two new computational methodologies that allow the simultaneous selection of the best regression models and the most interesting covariates to predict carcass and/or meat quality are developed. Then, a method of variable clustering is explained that is accurate in evaluating the interrelationships between different parameters of interest. Finally, some principles for the management of quality trade-offs are presented and the Meat Standards Australia model is discussed. The “Pareto front” is an interesting approach to deal jointly with the different sets of expectations and to propose a method that could optimize all expectations together.

## 1. Introduction

If a global rise in meat consumption is predicted worldwide, it can be seen that this increase does not concern all meat types [1]. For example, if we focus on the European Union, we can forecast an increase of consumption of poultry and pork meat (collectively referred to as white meat) with a simultaneous decrease of beef (referred to as red meat) consumption. Beef consumption in Europe has decreased in the past twenty years (−14%: from 17 kg of carcass equivalent in 2007 to 15 kg of carcass equivalent in 2018). In the same period, pork consumption has only decreased by 3% and poultry consumption has increased by 19% [2].

In the future, European beef consumption is expected to decline or stabilize, with a per capita consumption of 10.2 kg in 2028 (vs. 10.3 kg currently) [1]. In particular, consumption of fresh meat is expected to decrease, although this is likely to be offset by the increased use of meat products as ingredients in processed food [3].

In France, according to a recent survey made on 625 French consumers [4] the major reasons for the decline in beef consumption reflect the reasons previously identified in the literature [5,6]. Indeed, 37% of respondents associated the decline of beef consumption with too high variability in sensory quality, although also with a high price of beef. Mastering sensory quality is therefore a priority issue for the industry in order to stem the fall in consumption. Moreover, 35% of respondents were concerned about the possible health risks associated with beef consumption, combined with its suspected low nutritional value (i.e., high saturated fat level). Mastering nutritional quality is thus also a crucial issue for beef industry. Furthermore, the current exposure of meat production systems to the public has raised new social questions about environmental issues, human health, animal welfare and, indeed, whether animals should be slaughtered to produce human food [7].

Beyond consumers, it is also important to consider the whole meat supply chain, which includes all operators from farm to consumption. Thus, upstream of the supply chain, farmers are nowadays dependent on a fluctuating market where they have little control over prices. Good control of production costs is therefore their main lever to ensure a suitable margin. In cattle farming, as in most livestock farming, animal feed is the first item of expenditure (50% to 60% on average). It is therefore in the interests of farmers to select and breed efficient animals, that is to say, animals that are able to convert efficiently distributed feed into sales products. The individual efficiency of animals is therefore a key parameter for operators in the sector, particularly upstream operators [8].

Slaughterers and processors also have to overcome their own constraints in order to ensure the sustainability of their activity. Their main concern is the market structure and consumers’ demands (in relation to their expectations and consumption habits). The quality of carcasses is thus an important parameter for the meat sector, insofar as it determines the payment of the farmer, the remuneration of the intermediate link and the assurance of an optimized meat quality [8].

However, it is difficult to reach the expectations of all the stakeholders at the same time. So, in order to manage these conflicting requirements and the trade-offs needed, it is necessary to know precisely retailers’ and consumers’ expectations and to know how to assess carcass and meat quality according to these expectations.

With this in mind, the present work will review various methodological approaches that could allow the simultaneous control of expectations that are not always positively correlated, with the final aim being to better manage the trade-offs between different measures of quality in the beef chain sector. The objective here is therefore to propose methods and tools that can help in the evaluation and prediction of different types of quality. These methods will be illustrated for some carcass traits and beef tenderness.

## 2. What Are the Expectations Concerning Carcass and Meat Quality?

### 2.1. Carcass Quality Expectations

In Europe, the EUROP carcass classification system is based on global indicators, including the category of animals determined by gender (including steers), age, conformation, fat scores of carcasses, and hot carcass weight. A carcass is, however, a complex and heterogeneous entity, which, within the same EUROP classification, can comprise varying proportions of muscle and/or varying proportions of muscles with a higher or lower commercial value. The global characterization of carcasses by the EUROP system does not take this complexity into account. Indeed, the European grid is the simplest system in the world to grade carcasses. Unlike more complex systems as in Asia and North America (USDA), it does not take into account marbling, color or other traits recorded in the chiller [9].

Nevertheless, all grading systems (except the Australian system) are focused on carcasses, rather than on meat. The only grading system focused on meat is the Meat Standards Australia (MSA) grading scheme, which grades meat not carcasses. In addition, it includes traits measured not only in the slaughterhouse and in the chiller as other grading schemes do, but also traits recorded pre-slaughter and post-chiller (Table 1).

In order to improve the assessment of carcass quality obtained by the EUROP system, Monteils et al. (2017) [10] conducted a study based on a literature survey associated with a hierarchical structure according to the interests of different stakeholders in the meat chain. This work allowed the authors to propose a set of additional indicators taking into account their frequency of citations in the literature as well as their complementarity with the “historical” indicators of the EUROP grid [10]. These authors propose to complete the current set of EUROP indicators (based on carcass weight and sex, conformation and fat cover) with five other ones, namely: hindquarter weight, meat color, retail cut yield, rib eye area and marbling score.

In 2018 and 2019, a survey was conducted in various French slaughterhouses to determine, from the point of view of operators, which are the most important carcass quality indicators to consider (unpublished data). This work was carried out with 13 organizations marketing meat after slaughter, representative of the diversity of operators: four producer organizations or cooperatives, four slaughterhouses, three butchers and two breeders engaged in direct sales. In this study, we processed and analyzed data using an interface of R (R Core Team 2018), named IRaMuTeQ. Based on R software and python language, IRaMuTeQ extracted information from texts using descriptive statistics [11]. This survey did not intend to be exhaustive but rather to provide elements to validate the indicators proposed by [10] within the 1st French agricultural region: “Nouvelle-Aquitaine”. The objectives of the survey questionnaire were to:determine what are the expectations of the operators in the sector (slaughterers, butchers, direct sales farmers, cooperatives, etc.) regarding carcasses, according to their customers and market requirementsdetermine what constitutes an optimal quality carcass for different breeds and categories of animals according to the various stakeholdersestablish minimum quality thresholds to be reached for each of the specifications or each of the customer typeshighlight the criteria for assessing carcass quality

Outputs of this study were first the expectations of the various stakeholders in the meat sector in the region of “Nouvelle-Aquitaine” in terms of carcass quality for suckler cows. The main expectations were fairly homogeneous regardless of the outlet of the carcasses (supermarket shelves, cutting, parts, etc.) (Figure 1). Expectations were mainly oriented towards the muscular development of the carcass through performance such as yields (83% of citations) and conformation indicators (75% of citations). Operators specified that a minimum conformation was required for carcasses, which must also be as homogeneous as possible (in terms of weight, fat cover, etc.) in order to facilitate the processing of the carcass and preparation of meat cuts.

Marbling attracts the attention of stakeholders (58% of citations), since the consumer is becoming more and more interested in it. This descriptor is quite important, although the fattening state is rarely mentioned as a determining criterion. The operators indicate that the same fattening state can hide carcasses with a very different fat and marbling development. What is important for the stakeholders is to have carcasses of high quality (in terms of distribution of forequarter and hindquarter muscles, suitability for storage and maturation, etc.) but also to have ad hoc cutting of carcasses to allow marketing of small portions, suitable for self-service marketing. Meat color and tenderness (evaluated through handling and appreciation of the “meat grain” previously defined by [12] are also determinant descriptors (67% and 50% of citations respectively), especially for consumer satisfaction.

It is assumed that the addition of the five new indicators (hindquarter weight, meat color, retail cut yield, rib eye area and marbling score) proposed by [10] would complement the EUROP carcass classification in a beneficial way, permitting an improved meeting of the expectations of operators in the sector, and thus the expectations of final consumers.

As indicated earlier, additional indicators could improve assessment of carcass quality obtained with the EUROP system. However, the proposal to add new indicators to the current carcass rating parameters is also partly due to the fact that the EUROP rating does not in any way predict the potential eating quality [13] or nutritional quality.

### 2.2. Meat Quality Expectations

Meat eating quality refers to the characteristics of the product itself and includes especially sensory traits (e.g., tenderness, flavor, juiciness, overall liking), and healthiness, reviewed by [14,15]. Consumers’ expectations are thus of different kinds, but above all are very numerous. Indeed, a recent survey on 625 individuals indicates that the main reasons for the decline in beef consumption are: the too high price of beef, the possible health risks, the environmental impact of farming, the development of new consumption behaviors (less quantity and more quality), the lack of consistency in tenderness and taste, animal welfare concerns or the impact of health scandals [4]. Thus, many experts have made a distinction between intrinsic and extrinsic quality attributes of meat. The first refers to the product itself and includes, for instance, safety and health aspects, as well as sensory properties. The latter refers to traits more or less associated with the product, namely production system characteristics (including animal welfare, environmental aspects, and social considerations, for instance), as well as marketing variables. Each quality trait (intrinsic or extrinsic) is itself the aggregation of sub-criteria [16]. This gives rise to two questions: how to measure all these traits and how to aggregate them. Hocquette et al. [15] have suggested some ways to combine different quality criteria based on the existing literature. In the past, this aggregation was conducted by experts such as butchers who used to provide advice to consumers. However, this method is not exhaustive and also not consistent across butchers or meat experts. Another simple way is to define minimum thresholds: for instance, “this meat should contain a minimum of fat, a minimum of Poly Unsaturated Fatty Acids (PUFA) or of any type of vitamin”. This method is easy to understand and to implement but is a rough evaluation and requires routine measurements of the components of interest. A ranking system could also be defined to classify meat samples from the best (rank 1) to the worst (rank n), with a summation of the ranks with different traits. However, this is only a “relative” judgment, comparing alternatives among themselves, and not an “absolute” assessment. The best way is to convert quality traits into value scores (e.g., quantitative information on a common scale) which are then compounded, as done in the Australian system MSA (see below).

## 3. Modulation and Prediction of Quality Traits

### 3.1. Regression Models

We recently developed a new computational methodology that simultaneously selects the best regression model and the most interesting covariates [17] to predict carcass and/or meat quality. With such a method, we might predict one parameter using many variables. For instance, it could be possible to predict beef tenderness by various breeding factors and/or by animal performance.

In the modvarsel R-package, different models were tested (the linear regression, the PCR and the Slice Inverse regression, but also the random forest) but other ones (Support Vector Machine—[SVM], Ridge, Partial Least Squares—[PLSR]) could easily be practically implemented by the user. For each model, a number of selected variables has been reported. More precisely, this R package was used to select the proteins that could be considered predictive of meat tenderness among a pool of 21 candidate proteins assayed in *semitendinosus* muscle from 71 young bulls of the European ProSafeBeef project (Figure 2). The occurrence of each variable was calculated, leading to a ranking of variables according to their importance (Figure 3). By using the preselected settings, an algorithm proposes an optimal number of factors (in this case, an optimal number of proteins) to predict the variable of interest (in this case, tenderness), but it is also possible for the user to select the optimal number himself.

As a further development of the modvarsel R-package, we developed another statistical approach to select variables both in the group of co-variables and in the output parts which contain a group of variables to predict [17]. This method, called data-driven sparse partial least squares implemented in the ddsPLS R-package, may allow the prediction of several variables (for example, the tenderness scores of different muscles) by the same pool of factors (for example, breeding characteristics and/or animal performances). The use of a multi-block model allows for highlighting significant links between the tenderness and some co-variables. This approach made it possible to select and combine, respectively, three and four proteins capable of predicting the tenderness of the Triceps brachii and Gluteobiceps muscles. This confirms the interest and relevance of the method to accurately predict meat tenderness, as it appears that the combination of several variables (individually poorly correlated with tenderness) can provide a relevant prediction of tenderness.

### 3.2. Interrelations between the Various Quality Traits

Beyond the predictive models that allow prediction of a parameter of interest, such as tenderness, from a certain number of variables, it is important to determine how the different parameters of interest (such as carcass weight, carcass fatness or conformation, tenderness, flavor liking, juiciness, etc.) interact with each other. Indeed, knowing the interrelationships between these parameters is essential to propose breeding practices that allow the production of carcasses and meat with optimized qualities.

To determine how the different parameters of interest interact with each other, we recently proposed a methodological approach that could explore how to establish the links between different data sets, by using a variable clustering method [18] instead of the standard individual clustering as is usually done by principal component analysis [19].

This approach allows: (1)clarifying the interactions among different parameters of interest (for instance: animal performances, nutritional value, meat quality traits), and(2)assessing how to simultaneously control different parameters of interest that are not always positively correlated.

To test this method, data from 71 young bulls of the European ProSafeBeef project were used [14]. For each animal, 97 variables were collected and organized in three sets of data, characterizing animal efficiency and performance, nutritional value and sensory quality [20].

A clustering of variables was conducted using the R-package ClustOfVar. This is a dimension reduction method that can be a helpful tool to select variables. Indeed, each synthetic variable from ClustOfVar is a linear combination of a subset of original variables (whereas the principal components in principal component analysis are a linear combination of all the original variables). The clustering of variables allows the establishment of a total of 15 synthetic indexes (five per set of data). Then, a second clustering, realized on these 15 synthetic indexes, establishes the proximities between the three data sets.

The ClustOfVar approach used in this paper provided homogeneous groups of variables defined by a squared Pearson correlation [21]. This method, quite new in animal science, has already delivered obvious results in a number of areas such as the automobile industry or tourist cruise ships industry [21,22,23,24].

Our previous work [18] gave some new information to manage the trade-offs between different data sets (animal efficiency, nutritional value and meat quality traits in our case). This study aimed to replace the usual data mining analysis with a variable clustering approach. This method appears to be an effective tool to integrate various concepts in an optimized management of breeding factors and breeding practices.

After having analyzed the relationships between the different parameters of interest, there is a need to set up a method to manage the trade-offs between conflicting parameters to help the beef cattle industry to design specifications, taking into account the possible interactions between these different parameters of interest.

### 3.3. Trade-Off Management

Trade-offs are needed when a decision maker is in a situation when improving a criterion automatically implies a decrease in the score of another criterion. Such a definition implies that there exists a dependence between the two criteria.

In meat, such a situation often exists, for example in the case of tenderness and lipid content in a meat cut. There is a dependence since it was shown that higher lipid content induces a higher score for the evaluation of tenderness [25]. Therefore, if a meat brand wants to market a new product having a high tenderness and a low intramuscular fat content, the decision maker would have to make a trade-off between these two criteria.

However, there are other cases where improving a criterion does not automatically imply decreasing or increasing the other criterion. This means that the correlation between the two criteria is not strong (|*r*| << 1). Then, before choosing the trade-off between the two criteria, an optimal set has to be found. This set is also called the Pareto Front (PF) and is the solution of the multi-objective optimization (MOO) problem: (1)$$\underset{x}{\min}\left(f_{1}\left(x\right),\;\ldots ,\;f_{p}\left(x\right)\right)$$
where *x* is the decision vector holding q practice management parameters (for example, breed, feed intake, etc.), *f_j_* is a link function between the decision space and the j^e^ objective (for example, tenderness grade or lipids content), j∈{1,…,p} where p is the number of objective functions (Figure 4). The “min” operand is defined according the Pareto optimality based on the principle of domination, described in [26].

This stage was absolutely objective in the sense that no choices were made in the process of optimization. In contrast, the second stage seeking to perform the trade-off is subjective since the preferences of the decision maker are integrated in the process. Therefore, different final solutions for a given problem can occur if a decision maker integrates differing sets of preferences. Numerous techniques to deal with multicriteria decisions exist [27], varying in complexity but giving a hierarchical order in most of the cases to choose in a relative way the best trade-off amongst all the possible ones contained in the Pareto Front.

In the meat field, an approach using weight aggregation was used by [28] to find a trade-off between nutritional and sensory quality. This study reported that weight setting has a high influence on the final result and must be managed carefully. The study also observed that the model used was not taking into account all the complexity of the reality since compensation between criteria can result in selecting poor overall trade-off.

The underlying difficulty of the described process is to dispose of trusty link functions (the *f_j_* used later in the optimization search). Indeed, as discussed in the previous section, there is no analytical formula to model the links between animal practices and meat qualities. However, most of the time surrogate models can approximate them: (2)$$y_{j}=f_{j}\left(x\right)+\varepsilon_{j}$$
where *y_j_* is the variable of interest to model (tenderness, for example) and *ε_j_* is a random error term. The chosen statistical model (such as parametric, semiparametric or nonparametric model, with specific assumptions on *ε_j_*) will depend on each *y_j_* to be modeled. Consequently, the optimization search with a determinist approach (assuming that *ε_j_* is negligible) can lead to sub-optimal solutions. Fortunately, there exist more complex methods to deal with this uncertainty, like stochastic and robust optimization methods [29,30,31]. Our own approach to dealing with this uncertainty is promising but still in a validation stage.

### 3.4. Modelling Approaches Combining Different Quality Indicators including Their Interactions

The main pitfall concerning the management of trade-offs lies in the use of the threshold method. Indeed, whereas trade-off management allows for removing unsatisfactory beef samples based on the chosen thresholds, it can also discard a significant proportion of “good” samples. Indeed, in the area of eating quality (focused on tenderness, juiciness, flavor liking and overall liking), certain thresholds are often used. For instance, by using thresholds such as pH > 6, age > 18 months (for young bulls) or 30 months (for other bovines), fat score < 3, conformation < 0+, ageing time < 14 days (young bulls) or <7 days (other bovines), unsatisfactory grilled striploins will be removed from the market which is good, but 42% of striploins assessed as good quality by consumers would be discarded as well, which is wasteful (Figure 5; reviewed by [32]). A comparison of different eating quality systems showed that those which were based on such thresholds for whole animals/carcasses discarded a higher proportion of good quality meat compared to the Meat Standards Australia (MSA), which estimated quality on a cut-by-cut basis (Farmer, personal communication). This demonstrates the need for more complex quality prediction methods. The global modelling approaches which have been set up to solve this problem aim to provide breeders with some decision-making keys to adapt farm management in order to optimize carcass and meat quality. The “downstream management” consists of predicting eating quality based on a consideration of consumer responses and the use of these to determine the importance of production parameters rather than putting the emphasis on production and carcass conformation.

The MSA grading scheme (which has been developed to predict sensory quality of beef) is based on innovative statistical approaches using scores from the direct assessment of tenderness, juiciness, flavor and overall liking of cooked meat on a 0–100 scale by untrained consumers [33,34].

The first principle of the MSA system is to work with untrained consumers rather than expert panelists because they represent the “normal consumer/customer (who is not expert)” who buys meat without any training. The second principle is to combine these four traits to give a global quality score called “MQ4”. Statistical analysis, which is crucial in the modeling approach, has defined the best weighting coefficients which are roughly 30% for tenderness, flavor, and overall liking and 10% for juiciness. In addition, discriminant analysis was used to match these consumer sensory scores with the quality ranking of meat given by consumers. Consumers are also asked to class meat as unsatisfactory, good every day (3 *), better than every day (4 *) and premium (5 *). Furthermore, the values of the global MQ4 score defining the limits between each quality class are precisely calculated for each data set and are regularly refined: they are about 46 (between unsatisfactory and 3 *), 64 (between 3 * and 4 *) and 76 (between 4 * and 5 *) on a scale of 0 to 100.

The MSA model has been tested successfully in different countries (reviewed by Bonny et al., (2018) [35]) including France [36,37,38], South Korea [39], Northern Ireland [40,41,42,43], the USA [44,45], Japan [46], Ireland [47], South Africa [48], New Zealand [49], and Poland [50,51]. The general conclusion is that the MSA methodology is relevant in all these countries, indicating that the MSA model is likely to be generally applicable. However, the relative weighting coefficients for tenderness, flavor liking, juiciness, and overall liking in the optimal calculation of the MQ4 score vary slightly between countries, and the optimal limits between quality classes can be refined for each country or for each group of consumers.

For quality prediction, a mathematical model has been built to predict the eating quality of beef for each “muscle × cooking method” combination. This model was constructed from a large database of consumer tests using a standard protocol. A dozen parameters having a statistically significant effect on eating quality, such as traits characterizing animals, pre-slaughter and slaughter conditions, meat, and post-mortem events are considered in the model as well as the interactions between them.

In practice, the slaughterhouse is the backbone of the system. A specific grader, who is accredited after training and who is periodically trained, grades the carcasses for marbling, fat and meat color, ossification, which is an indicator of age and more precisely of physiological maturity. Temperature and pH are also recorded. The MSA model then predicts the overall quality score of MQ4 on a scale of 0 to 100, as well the quality class for each piece of meat associated with a cooking method and a specific ageing time. All factors related to the animal, to its carcass and its meat are recorded and included in the model for prediction. So far, marbling, ageing time and the carcass hanging method were not taken into account in France. However, a French private company (Beauvallet/CV Plainemaison) has launched a new premium beef breed called OR ROUGE based on these traits.

Research on the application of the substantial data gained using MSA protocols in Europe and elsewhere will continue within the activities of the International Meat Research 3G Foundation. This Foundation was launched in 2017 under the auspices of the United Nations Economic Commission for Europe (UNECE) [52,53]. The Meat3G foundation is likely to develop a new MSA-like model to predict eating quality of beef across countries and based on data gathered in different countries. The standard protocols of carcass grading based on the MSA protocols have been approved by the United Nations Economic Commission for Europe (UNECE).

As previously detailed, many models were recently developed in order to predict each quality trait and to evaluate the possible trade-offs that could be accepted in order to satisfy all the operators of the beef chain at the same time. In order to summarize our subject, it is possible to group together in Table 2 the main objectives, advantages and disadvantages of the statistical methods developed in this paper.

## 4. Conclusions

Consumer satisfaction when eating beef is a complex response based on subjective and emotional assessments. Safety and healthiness are very important in addition to taste and convenience for consumers, but some other parameters, such as yield and conformation, are really important for breeders. A variety of modelling approaches have been tested to assess and predict carcass and meat quality traits. Amongst these, the “Pareto front” approach proves to be an interesting method to optimize all stakeholder expectations. The MSA model has been proven to deliver improved eating quality when tested in numerous countries around the world. The first step will be to look for the set of non-dominant solutions (i.e., possible compromises from which the decision-maker has to choose). It is, therefore, now necessary to draw up a list of quality parameters sought to determine their limits and the minimum/maximum acceptable values for each parameter. It will also be necessary to study the existence of possible combinations between the different expectations. Then, the list of criteria to be optimized will have to be drawn up and prioritized by experts in order to know how to satisfy the expectations of the different stakeholders of the sector. These different aspects are currently in progress.

## Figures and Tables

**Figure 1 foods-09-00525-f001:**
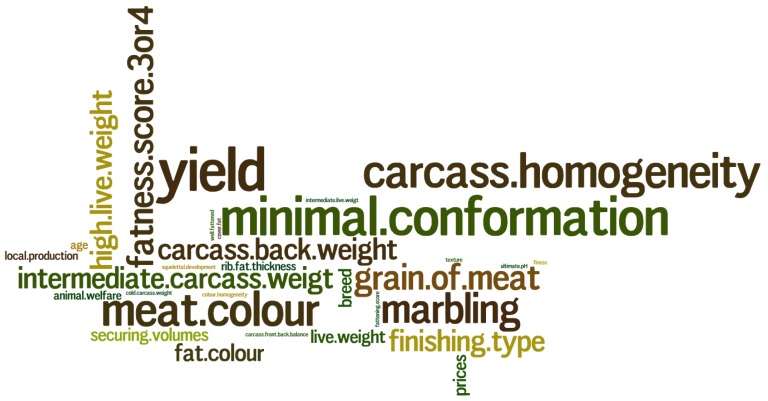
Main expectations expressed by the 13 operators in the sector surveyed. (The size of the expectations was proportional to its percentage of citation. For instance, yield was recorded in 83% of citations; minimal conformation 75%; carcass homogeneity and meat color: 67%; marbling: 58%; high live weight and fatness score 3 or 4: 50%; the other expectations were recorded in less than 50% of citations).

**Figure 2 foods-09-00525-f002:**
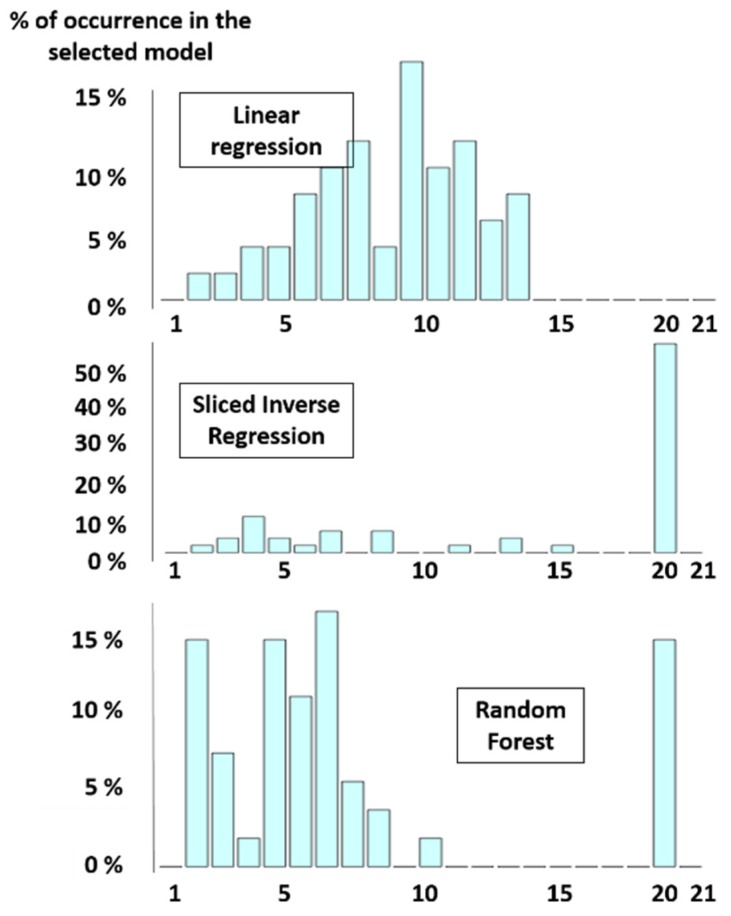
Number of selected variables for three different models (in this example, 21 factors were used to predict the variable of interest, which is tenderness) (adapted from [17]). In this example, 17% of the linear regression (LR) models use 10 variables and all of the LR models use less than 14 variables. On the contrary, 60% of the slice inverse regression (SIR) models use 20 variables out of 21 to predict the parameter of interest. About 15% of the Random Forest (RF) models use 20 variables out of 21 to predict tenderness, whereas, more than 80% of the RF models use between two and eight variables to predict this parameter.

**Figure 3 foods-09-00525-f003:**
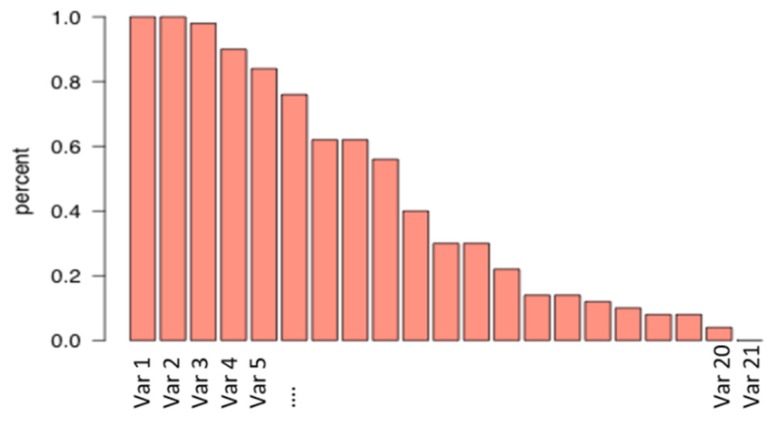
Occurrence of each variable in the selected models (adapted from [17])). In this example, the variables 1 and 2 are selected in 100% of models. The variable 21 is selected in less than 1% of the models and is therefore not very informative and not necessary in the model for predicting the parameter of interest (which is tenderness in this example).

**Figure 4 foods-09-00525-f004:**
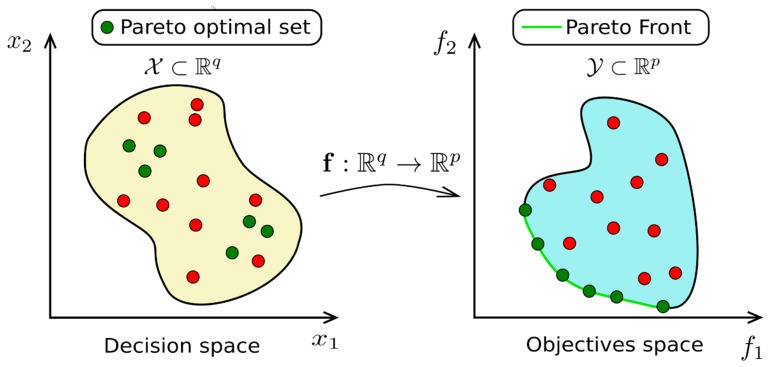
Scheme of the Pareto front and its optimal set (adapted from [26]).

**Figure 5 foods-09-00525-f005:**
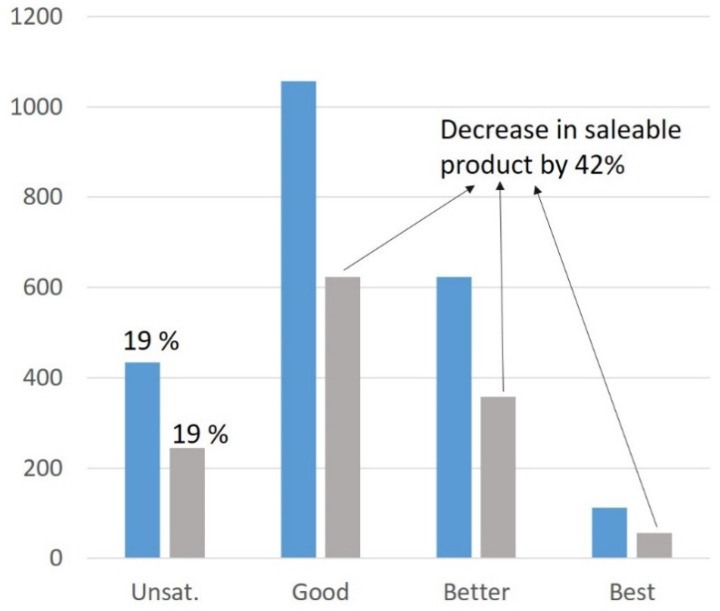
Number of samples in each quality category for grilles striploins (blue) and after applying threshold criteria (grey). The % failure is displayed above the fall column [32]. Threshold criteria are pH (>6), age (>30 months for females and steers, >18 months for young bulls), conformation (<0+), fat score (<3), aging length (<7 days for females and steers; <14 days for young bulls).

**Table 1 foods-09-00525-t001:** Major beef carcass classification systems implemented throughout the world (adapted from [9]).

**Country**	**Europe**	**S. Africa**	Canada	Japan	S. Korea	USA	Australia
Scheme	EUROP	S. Africa	Canada	JMGA	Korea	USDA	MSA
Grading unit	Carcass						Cut
**Pre slaughter factors**							HGP implants & Bos Indicus
Slaughter-floor	Carcass weight and sex						
	Conformation	Dentition	Conformation				Electrical stimulation
	Fat cover	ribfat					Hang
Chiller			Marbling score				
			Meat color				
			Fat color and fat thickness			Ossification score	
			Texture	Eye muscle area			Fat thickness
				Meat brightness	Texture	Meat texture	Hump height
				Fat luster	Firmness	Rib fat	Ultimate pH
				Fat texture	Lean maturity	Kidney fat	
				Fat firmness		Perirenal fat	
				Rib thickness			
Post chiller							Ageing time
							Cooking method

**Table 2 foods-09-00525-t002:** Objectives, advantages and disadvantages of the statistical approaches developed in the present paper.

	Objective	Advantages	Disadvantages
Regression model	Estimation of model to explain a single parameter by many covariates.	Easy model interpretability thanks to a parametric modeling. Easy prediction method.	Linear model. Single parameter modeling. Single block of covariates. Need of a sample size greater than the number of covariates.
modvarsel R-package	Regression model benchmark and variable selection	Wide choice between several parametric, semi-parametric or non-parametric regression models. Ranking of variables according to their importance allowing simple selection of variables. Easy prediction method. Easy to use.	Computational burden. Single parameter modeling. Single block of covariates.
ddsPLS R-package	Modeling and selection of variables to predict and of traits to be predicted	Prediction of several parameters by the same pool of factors. Multi-block approach: various blocks of covariates and one block of parameters to explain. Adapted for a small sample size much lower than the number of covariates.	Linear model. Only numerical covariates and response blocks. Interpretation of the outputs slightly more technical.
ClustOfVar R-package	Approach providing a clustering of variables based on their correlations	Identification of interactions/links allowing dimensional reduction of variables-via the scores (synthetic variables) associated with each cluster. Easy interpretability of the scores. Method adapted to quantitative and qualitative variables. Hierarchical clustering or not. Easy to use.	Possible correlation between the cluster scores. Only linear correlations (or correlation ratios) taken into account.
Trade-off management	Decision-making methodology for a compromise between different quality objectives.	Integration of priority preference of the decision maker. Easy to use.	Need of a big amount of data to be accurate. Discard of unsatisfactory but also relevant samples
Meat Standards Australia (MSA)	Decision-making methodology based on the combination of different sensory quality traits	Inclusion in the model of different variables and of their interactions. Easy interpretability of the scores. Continuous improvement of the model. Method already implemented in the Australian beef industry with success.	Need of a big amount of data to be accurate.

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
