# Peer review of "Various Statistical Approaches to Assess and Predict Carcass and Meat Quality Traits"

_foods, 2020, doi:10.3390/foods9040525_

Round 1

Reviewer 1 Report

General comments

The manuscript is very interesting for everyone associated with beef industry. It provides a competent knowledge about all the methods /systems allowing to assess/ predict the quality of beef carcasses and meat. However, before it can be published, it requires some minor corrections. The language has to be improved – not only the English correction is needed. Some sentences/ terms are of ‘popular science’ nature, and should be changed.

Also, the whole Conclusions should be rearranged/ corrected. See in the specific comments.

Specific comments

Title

Proposal of various recent statistical approaches to assess and predict carcass and meat quality traits’, should be changed to: ‘Various statistical approaches to assess and predict carcass and meat quality traits – a review’

Abstract

Line 20. ‘actors’ – did you really want to use this word here?

Line 29. MSA – this abbreviation needs to be explained when it appears the first time

Introduction

Line 36. ‘(white meat)’ – please changed to ‘(collectively referred to as white meat)’

Line 37. ‘(red meat)’ - Please change to ‘(referred to as red meat)’.

Line 51-51. ‘Mastering sensory and nutritional quality is therefore a priority and a crucial issue for the industry in order to stem the fall in consumption’, while before stating this, you mention about the consumer concerns about health risks connected with beef consumption.

So maybe it’s not just about ‘sensory and nutritional quality’ in the meat industry?

Main text

Line 86. ‘non-recoverable’ – probably you meant ‘non-carcass parts’ or non-edible animal by-products? You should use a more precise terms.

Line 93. ‘on meat and this is a problem because consumers, of course, do not eat carcasses, but pieces of meat’ – this sentence should be converted, it’s a language of a ‘popular science’ rather than a serious scientific journal.

Line 140. ‘actors’ ??

Line 112. In 2018 and 2019, a survey was conducted …’ – a reference should be provided

Lines 160-161. ‘Meat quality refers to the characteristics of the product itself and includes especially sensory traits (e.g. tenderness, flavor, juiciness, overall liking), and healthiness, reviewed by (Hocquette et al., 2014, 2012)’ – it’s a very narrow definition. Meat quality can analysed from the technological point of view (usefulness in processing) or meat safety (microbiological quality).

Or maybe you are describing eating quality of meat? In this case, the definition is more suitable.

Line 167. ‘environmental quality, social (ethical) ‘quality and economic quality’ – is it proper to use these terms in respect of meat?

Line 193 – 226. A lot of research you mention refers to less popular muscle (less valuable from the culinary point of view): Semitendinosus, triceps brachii, gluteobiceps.. Are there any research on predicting quality traits (ex. Tenderness ) of muscles like longissimus thoracis et lumborum, or semimembranosus?

Conclusions – need a general improvement

The summary of all described models/methods could be placed in a table. But it should be presented in the main text, not in conclusions. Maybe you should create a separate section to focus all the pros and cons of different methods in one place.

In the Conclusions you can only ‘conclude’ which method was the best/ worst, and what should be improved in the future and why.

Figures

Figure 1. It should be presented in other form – maybe a percentage of main expectations in the sector? In the current form Figure one is unclear and lacks a scientific character.

Figure 2 and 3. If these figures are adapted from an article (even if it was written by the same author), this should be stated.

Author Response

Dear Editor

We thank the referees for the careful review of our manuscript. Please, find here below answers to all the points raised by the referees with corresponding changes in the paper.

-Reviewer 1

The manuscript is very interesting for everyone associated with beef industry. It provides a competent knowledge about all the methods /systems allowing to assess/ predict the quality of beef carcasses and meat.

However, before it can be published, it requires some minor corrections. The language has to be improved – not only the English correction is needed. Some sentences/ terms are of ‘popular science’ nature, and should be changed. Also, the whole Conclusions should be rearranged/ corrected. See in the specific comments.

The use of English has been reviewed throughout the paper and language corrections/modifications are added to this revised version, according to referee’s suggestions.

Specific comments

Title

Proposal of various recent statistical approaches to assess and predict carcass and meat quality traits’, should be changed to: ‘Various statistical approaches to assess and predict carcass and meat quality traits – a review’

We changed the title as proposed. Nevertheless, we choose not to indicated “a review” as we haven’t really done a review but only a proposal of the recent approaches

Abstract

Line 20. ‘actors’ – did you really want to use this word here?

Modified: stakeholders instead of actors

Line 29. MSA – this abbreviation needs to be explained when it appears the first time

Modified

Introduction

Line 36. ‘(white meat)’ – please changed to ‘(collectively referred to as white meat)’

Modified

Line 37. ‘(red meat)’ - Please change to ‘(referred to as red meat)’.

Modified

Line 51-51. ‘Mastering sensory and nutritional quality is therefore a priority and a crucial issue for the industry in order to stem the fall in consumption’, while before stating this, you mention about the consumer concerns about health risks connected with beef consumption.

So maybe it’s not just about ‘sensory and nutritional quality’ in the meat industry?

Modified in order to be clearer

Line 86. ‘non-recoverable’ – probably you meant ‘non-carcass parts’ or non-edible animal by-products? You should use a more precise terms.

‘non recoverable” was removed

Line 93. ‘on meat and this is a problem because consumers, of course, do not eat carcasses, but pieces of meat’ – this sentence should be converted, it’s a language of a ‘popular science’ rather than a serious scientific journal.

We removed this part of the sentence.

Line 140. ‘actors’ ??

Modified: stakeholders instead of actors

Line 112. In 2018 and 2019, a survey was conducted …’ – a reference should be provided

We indicated that there are unpublished data

Lines 160-161. ‘Meat quality refers to the characteristics of the product itself and includes especially sensory traits (e.g. tenderness, flavor, juiciness, overall liking), and healthiness, reviewed by (Hocquette et al., 2014, 2012)’ – it’s a very narrow definition. Meat quality can analysed from the technological point of view (usefulness in processing) or meat safety (microbiological quality).

Or maybe you are describing eating quality of meat? In this case, the definition is more suitable.

Modified : we indicated “eating quality of meat”

Line 167. ‘environmental quality, social (ethical) ‘quality and economic quality’ – is it proper to use these terms in respect of meat?

Many experts have made a distinction between intrinsic and extrinsic quality attributes of meat. The first refers to the product itself and includes for instance, (i) safety and health aspects, (ii) sensory properties (e.g. texture and flavour) and shelf life, (iii) chemical and nutritional attributes and (iv) reliability and convenience. The latter refers to traits more or less associated with the product, namely (i) production system characteristics (from the animals to processing stages including animal welfare, environmental aspects, and social considerations for instance), and (ii) marketing variables (including price, brand name, distribution, origin, packaging, labelling, and traceability) (LUNING, P.A., MARCELIS, W.J., JONGEN, W.M.F., 2002. Food quality management. A technico-managerial approach. Wageningen Pers., Wageningen, The Netherlands; GRUNERT, K.G., BREDAHL, L., BRUNSO, K., 2004. Consumer perception of meat quality and implications for product development in the meat sector - a review. Meat Sci. 66:259-272.).

Line 193 – 226. A lot of research you mention refers to less popular muscle (less valuable from the culinary point of view): Semitendinosus, triceps brachii, gluteobiceps.. Are there any research on predicting quality traits (ex. Tenderness ) of muscles like longissimus thoracis et lumborum, or semimembranosus?

The experiments used here have focused on muscles that are certainly less common than the longissimus thoracis, but which have the advantage of having extreme (especially muscular) characteristics. 

Conclusions – need a general improvement

The summary of all described models/methods could be placed in a table. But it should be presented in the main text, not in conclusions. Maybe you should create a separate section to focus all the pros and cons of different methods in one place.

In the Conclusions you can only ‘conclude’ which method was the best/ worst, and what should be improved in the future and why.

Consumer satisfaction when eating beef is a complex response based on subjective and emotional assessments. Safety and healthiness are very important in addition to taste and convenience, but some other parameters are really important for breeders (farmers?).

Among many models, the Meat Standards Australia grading scheme or the "Pareto front" approach proved to be interesting methods to optimize together the sensory traits or all consumer’s expectations respectively.

In the latter case, the first step will be to look for a set of open solutions (i.e. possible compromises from which the decision-maker has to choose). It is therefore now necessary to list quality parameters of interests, to determine their limits and the minimum/maximum acceptable values for each parameter. It will be also necessary to highlight the existence of potential combinations between the different expectations. Then, the criteria to be optimized will have to be listed and prioritized by experts, in order to know how to satisfy the expectations of the different stakeholders of the sector. These different aspects are currently in progress.

Figures

Figure 1. It should be presented in other form – maybe a percentage of main expectations in the sector? In the current form Figure one is unclear and lacks a scientific character.

We added this information in the comment to the figure but also in the text

Figure 2 and 3. If these figures are adapted from an article (even if it was written by the same author), this should be stated.

The reference was added

-Reviewer 2

From a European / French point of view, the paper describes and comments on different meat grading systems around the world.

Computer programs developed by the authors (in R) are referred to, and some overall results presented.

In the last part of the paper the MSA system of Australia is rather thoroughly described, without clear link to the previously mentioned computer programs.

As an overview of the field of meat grading and possibilities for the future research and development, I find the paper interesting. But for any application of the methods mentioned a reading of more detailed papers would be required.

The description of the MSA system is now more focused on the statistical and the modelling approach. There is indeed no real link to the previously mentioned computer programs. It is now presented as an additional statistical model.

We agree that for any application of the methods mentioned a reading of more detailed papers would be required. However, the aim of this manuscript is to briefly present the advantages and limits of each methods, to compare them, not to describe them in details.

Some specific comments to the manuscript

Are the 71 bulls mentioned in line 247 the same 71 as the ones mentioned earlier in the manuscript? This should be clearer.

Yes, these are the same. We added this information in the text

What is 'cruise industry' in line 260? Tourist cruise ships?

Modified

Examples of language that could be adjusted (there are many more):
20 The beef industry is organized around different actors, each with its own
expectations,
->
20 The beef industry is organized around different actors, each with their
own expectations,

Modified, sorry for this careless mistake.

27 developed. Then, the method of variable clustering
->
27 developed. Then, a method of variable clustering

Modified, sorry for this careless mistake.

218 In the continuity of the modvarsel R-package,
->
218 As a step further from the modvarsel R-package,

Modified, sorry for this careless mistake.0

307 all the possible ones contains in the Pareto Front.
->
307 all the possible ones contained in the Pareto Front.

Modified, sorry for this careless mistake.

312 criteria can result in selecting poor trade-off on overall.
312 criteria can result in selecting poor overall trade-off.

Modified, sorry for this careless mistake.

313 The underlying difficulty of the described process is to dispose of
trusty link functions (the ?? use later
->
313 The underlying difficulty of the described process is to dispose of
trusty link functions (the ?? used later

Modified, sorry for this careless mistake.

In 334 there is a space between < and 7, but not in the '<14 days' in 333

Modified, sorry for this careless mistake.

Unclear phrase 342-344:
When predicting eating quality is to consider first consumer responses and
use these to determine the importance of production traits rather than
putting the emphasis on production traits.

The sentence was modified in order to be clearer

We sincerely hope that our manuscript will be accepted for publication after these changes.

Best regards

Marie-Pierre Ellies-Oury on behalf of all authors

Reviewer 2 Report

From a European / French point of view, the paper describes and comments on different meat grading systems around the world.

Computer programs developed by the authors (in R) are referred to, and some overall results presented.

In the last part of the paper the MSA system of Australia is rather thoroughly described, without clear link to the previously mentioned computer programs.

As an overview of the field of meat grading and possibilities for the future research and development, I find the paper interesting. But for any application of the methods mentioned a reading of more detailed papers would be required.

Some specific comments to the manuscript

Are the 71 bulls mentioned in line 247 the same 71 as the ones mentioned
earlier in the manuscript? This should be clearer.

What is 'cruise industry' in line 260? Tourist cruise ships?

Examples of language that could be adjusted (there are many more):
20 The beef industry is organized around different actors, each with its own
expectations,
->
20 The beef industry is organized around different actors, each with their
own expectations,

27 developed. Then, the method of variable clustering
->
27 developed. Then, a method of variable clustering

218 In the continuity of the modvarsel R-package,
->
218 As a step further from the modvarsel R-package,

307 all the possible ones contains in the Pareto Front.
->
307 all the possible ones contained in the Pareto Front.

312 criteria can result in selecting poor trade-off on overall.
312 criteria can result in selecting poor overall trade-off.

313 The underlying difficulty of the described process is to dispose of
trusty link functions (the ?? use later
->
313 The underlying difficulty of the described process is to dispose of
trusty link functions (the ?? used later

In 334 there is a space between < and 7, but not in the '<14 days' in 333

Unclear phrase 342-344:
When predicting eating quality is to consider first consumer responses and
use these to determine the importance of production traits rather than
putting the emphasis on production traits.

Author Response

(The authors gave the same response as above.)
